



# Bioaerosols and atmospheric ice nuclei in a Mediterranean dryland: Community changes related to rainfall

Kai Tang [1,*], Beatriz Sánchez-Parra [1,2,*], Petya Yordanova [1,*], Jörn Wehking [1,a], Anna T. Backes [1], Daniel A. Pickersgill [1], Stefanie Maier [2], Jean Sciare [3], Ulrich Pöschl [1], Bettina Weber [1,2], and Janine Fröhlich-Nowoisky [1]

[1]Multiphase Chemistry Department, Max Planck Institute for Chemistry, P.O. Box 3060, 55020 Mainz, Germany
[2]Institute of Biology, University of Graz, Holteigasse 6, 8010 Graz, Austria
[3]Climate and Atmosphere Research Center, The Cyprus Institute, 2121 Nicosia, Cyprus
[a]now at: HygCen Austria GmbH, Werksgelände 28 5500 Bischofshofen, Austria
[*]These authors contributed equally to this work.

**Correspondence:** Janine Fröhlich-Nowoisky (j.frohlich@mpic.de)

**Abstract.**

Certain biological particles are highly efficient ice nuclei (IN), but the actual contribution of bioparticles to the pool of atmospheric IN and their relation to precipitation are not well characterized. We investigated the composition of bioaerosols, ice nucleation activity, and the effect of rainfall by metagenomic sequencing and freezing experiments of aerosol samples

collected during the INUIT 2016 campaign in a rural dryland on the Eastern Mediterranean island Cyprus. Taxonomic analysis showed community changes related to rainfall. For the rain-affected samples, we found higher read proportions of fungi, in particular of Agaricomycetes, which are a class of fungi actively discharging their spores into the atmosphere in response to humidity changes. In contrast, the read proportions of bacteria were reduced, indicating an effective removal of bacteria by precipitation. Freezing experiments showed that the IN population in the investigated samples was influenced by both rainfall

and dust events. For example, filtration and heat treatment of the samples collected during and immediately after rainfall yielded enhanced fractions of heat-sensitive IN in the size ranges larger than 5 μm and smaller than 0.1 μm, which were likely of biological origin (entire bioparticles and soluble macromolecular bio-IN). In contrast, samples collected in periods with dust events were dominated by heat-resistant IN active at lower temperatures, most likely mineral dust. The DNA analysis revealed low numbers of reads related to microorganisms that are known to be IN-active. This may reflect unknown sources

of atmospheric bio-IN as well as the presence of cell-free IN macromolecules that do not contain DNA, in particular for sizes < 0.1 μm. The observed effects of rainfall on the composition of atmospheric bioaerosols and IN may influence the hydrological cycle (bioprecipitation cycle) as well as the health effects of air particulate matter (pathogens, allergens).

## 1 Introduction

Primary biological aerosol particles, or short bioaerosols, are a subset of atmospheric aerosol particles directly emitted from

the biosphere into the atmpshere. They comprise living and dead microorganisms (e.g., bacteria), dispersal units (e.g., fungal spores, plant pollen), fragments (e.g., plant debris), and viruses (Després et al., 2012; Whon et al., 2012; Fröhlich-Nowoisky



et al., 2016; Smets et al., 2016; Yahya et al., 2019; Pöhlker et al., 2021). Bioaerosols can serve as nuclei for cloud droplets and ice crystals influencing the formation of precipitation, and some of them are important allergens and pathogens (Brown and Hovmøller, 2002; D'Amato et al., 2007; Andreae and Rosenfeld, 2008; Steiner et al., 2015; Müller-Germann et al., 2017;

Fröhlich-Nowoisky et al., 2016; Lang-Yona et al., 2018; Huang et al., 2021). The abundance and composition of bioaerosols are determined by local emission sources, meteorological conditions, and long-range transport (Jones and Harrison, 2004; Franzetti et al., 2011; Bowers et al., 2012, 2013; Gandolfi et al., 2013; Cao et al., 2014; Archer et al., 2020).

Rainfall and the associated increase of humidity can lead to increased emission of biological aerosol particles (Heo et al., 2014; Wright et al., 2014; Rathnayake et al., 2017). For example, raindrops hitting leaf or soil surfaces can disperse bacteria

and other bioparticles by splashing (Hirst, 1953; Madden, 1997; Gilet and Bourouiba, 2014; Perryman et al., 2014; Kim et al., 2019; Gregory et al., 1959; Joung et al., 2017). Furthermore, the increase of relative humidity can trigger the emission of fungal spores by active discharge mechanisms (Zoberi, 1964; Lacey, 1996; Elbert et al., 2007), and pollen grains can rupture under moist conditions releasing sub-pollen particles (Taylor et al., 2002, 2004; Taylor and Jonsson, 2004; Miguel et al., 2006; Steiner et al., 2015).

The release of bioaerosols, which can serve as cloud condensation nuclei (CCN) or ice nuclei (IN), can in turn influence the evolution of clouds and precipitation, thus closing a feedback cycle known as bioprecipitation (Sands et al., 1982; Möhler et al., 2007; Morris et al., 2014; Steiner et al., 2015; Fröhlich-Nowoisky et al., 2016). Especially over vegetated regions, in marine environments, or under remote conditions, bioparticles might represent a significant fraction of CCN and IN (Andreae and Rosenfeld, 2008; Pöschl et al., 2010; Pöhlker et al., 2012; Burrows et al., 2013; Wilson et al., 2015). As IN of biological origin

(bio-IN) can trigger freezing at higher temperatures than mineral IN, they are expected to play a role at cloud temperatures $> -15\,°C$ (Hoose and Möhler, 2012; Fröhlich-Nowoisky et al., 2016; Coluzza et al., 2017; DeMott and Prenni, 2010).

Increased concentrations of atmospheric IN during and after rainfall have been linked to bioaerosols (e.g., Huffman et al., 2013; Prenni et al., 2013; Schumacher et al., 2013; Tobo et al., 2013; Bigg et al., 2015; Hara et al., 2016b), but the sources and composition of bioparticles, their contribution to the pool of atmospheric IN, and their relation to rainfall are still not well

characterized. In this study, we investigated the effects of a short period with rain events on bioaerosol and atmospheric IN composition in a rural dryland. Aerosol samples of boundary layer air were collected on the Mediterranean island Cyprus in April 2016. The taxonomic composition of bioaerosols was determined by shotgun metagenomic sequencing and analysis. Ice nuclei were measured using the high-throughput Twin-plate Ice Nucleation Assay (TINA). Additional filtration and heat treatment experiments were performed to narrow down the IN size ranges and to investigate the heat sensitivity of the IN.

## 2 Materials and methods

### 2.1 Aerosol sampling

Aerosol sampling was performed during the INUIT-BACCHUS-ACTRIS campaign in April 2016 at the Cyprus Atmospheric Observatory, which is a rural background station that operates under the co-operative program for monitoring and evaluation of the long-range transmission of air pollutants in Europe (EMEP) and the European Research Infrastructure for the observation





of Aerosols, Clouds and Trace gases Research Infrastructure (ACTRIS) networks, while at the same time, it is a designated
regional Global Atmospheric Watch (GAW) station. The station (35.038692°N, 33.057850°E) is located close to the villages
of Agia Marina (∼630 inhabitants) and Xyliatos (∼150 inhabitants) and has an elevation of 532 m above sea level. The site
lies at the northeastern foothills of the Troodos Mountains. The nearest main urban agglomeration is at least 35 km away.
The vegetation at the measurement site is dominated by evergreen scrubs and small trees, blending at higher elevations of the

Troodos Mountains into oak and pine forests (Fall, 2012).

Total suspended particle samples were collected on glass fiber filters (Type MN-85/90BF, 150 mm diameter, retention ca-
pacity of 0.5 µm; Macherey-Nagel GmbH & Co. KG, Germany) with a High-Volume Aerosol Sampler DHA-80 (DIGITEL
Elektronik AG, Hegnau, Switzerland; volumetric flow rate 1000 L min$^{-1}$, sampling time 24 h) positioned at the roof top of the
Cyprus Atmospheric Observatory station (∼5 m above the ground). Blank samples were taken at regular intervals (∼3 day

intervals) to detect potential contamination by the aerosol sampler or by filter mounting. For this, filters were mounted in the
sampler as for regular sampling, but the sampler was either not turned on at all (mounting blank) or only for 5 s (start-up blank).

Prior to sampling, all glass fiber filters were pre-baked at 330 °C for 8 to 10 h in aluminum bags to remove any biological
material. The filter holders were disinfected with Bacillol® AF (Bode Chemie GmbH, Hamburg, Germany) before use. After
sampling, the loaded filters were removed from the filter holders, placed into baked (330 °C, 8 to 10 h) aluminum bags, and

stored at -20 °C, and at -80 °C after international transport on dry ice. The aerosol filter samples were cut into 16 equal
pieces using a self-made cutting tool, made of gilded sharp blades stuck to a round aluminum frame (140 mm diameter and
approximately 1 cm thickness). Prior to cutting, the cutting tool was sterilized by dipping it into ethanol (96 %) and flaming.
The filter sample aliquots were transferred into sterile 50 mL tubes (Greiner Bio-One, Kremsmünster, Austria) and stored at
-80 °C until analysis. To avoid contamination, filter handling and processing was carried out under sterile conditions on a clean

bench.

Based on the monitored environmental conditions as detailed below, nine aerosol samples were selected for metagenomic
sequencing and ice nucleation measurements. A list of the investigated aerosol samples with sampling details is given in
Table A1.

## 2.2  Meteorological conditions

The meteorological conditions during the sampling campaign in April 2016 were measured at the Cyprus Atmospheric Ob-
servatory by the Department of Labour Inspection of Cyprus. The data, comprised of hourly mean values for temperature,
relative humidity, rain fall, wind speed, and regulatory mass concentrations for PM$_{10}$ and PM$_{2.5}$ were imported into R version
3.6.3; (R-Development-Core-Team, 2011) and the Coordinated Universal Time (UTC) timestamps were converted to the local
Eastern European Time (EET: UTC +2:00).

Figure A1 shows the meteorological conditions and aerosol mass concentrations during collection of the aerosol samples.
Temperature and relative humidity (RH) exhibited some variation throughout the campaign. Noticeable is a strong increase of
RH on 9 April 2016, followed by a period of high RH (up to 80 %) and a temperature drop until 14 April 2016 corresponding



to samples CY19 –CY26. Except for this 5-day period, the daily temperature peaks were in the range of 25 to 30 °C. A period of higher RH was again registered starting on 25 April 2016 (CY48).

April 2016 was mainly dry, with only three rain events. One rain event was registered on 12 April during the last 3 h of collection of sample CY20, and two small rain events occurred during collection of CY23 on 13 April. Increased PM concentrations, corresponding to dust events (Schrod et al., 2017)), were measured from 9 April to 12 April (CY18 and CY19) and from 15 to 18 April (CY31 and CY32).

## 2.3   DNA extraction

Two filter sample aliquots from opposite sides of each air filter sample were extracted with the ChargeSwitch[TM] Forensic DNA Purification Kit (Invitrogen Corporation Thermo Fisher Scientific Inc., Waltham, MA, USA) according to the manufacturer's instructions with the following modifications: incubation at 55 °C for 60 min and elution of DNA with 100 μL elution buffer. Moreover, the DynaMag [TM]-2 instead of MagnaRack[TM] was used. DNA extracts of the two filter pieces of each sample were pooled and purified with the PowerClean Pro DNA Clean-Up Kit (MO BIO Laboratories, Inc. QIAGEN, Hilden, Germany)
according to manufacturer's instructions. To detect potential contamination, four mounting blanks and four start-up blanks were included in the DNA extraction. Additionally, four baked filters were included as filter blanks, and four extractions without any filter served as extraction blanks. DNA concentrations were determined with the DeNovix dsDNA Ultra High Sensitivity Assay (DeNovix Inc. DE, USA) and the Qubit[®] 3.0 Fluorometer (Thermo Fisher Scientific, USA). DNA was not detected in the mounting, start-up, filter, and extraction blanks, indicating that no contamination occurred during sample handling and
extraction.

## 2.4   Shotgun metagenome sequencing

From each DNA extract, one nanogram DNA was sequenced on Illumina HiSeq 3000 (output $2 \times 150$ bp paired-end sequencing reads) and Illumina HiSeq 2500 sequencers (output $2 \times 150$ bp paired-end sequencing reads) at the Max Planck Genome Centre in Cologne, Germany. Illumina TruSeq adapters were used for the sequencing library. The data output per run was 17.5 GB
reads. In total $4.14 \times 10^8$ reads were obtained from all samples after quality filtering, with an average of $4.6 \times 10^7$ reads per sample.

## 2.5   Taxonomic classification

To identify ribosomal small subunit (SSU; 16S/18S) and large subunit (LSU; 23S/28S) reads and to assign taxonomy and operational taxonomic units (OTUs), the MGnify pipeline 4.1 with MAPseq (1.2.2) and the SILVA database (SSU/LSU version
132) remapped to an eight-level-taxonomy was used (Mitchell et al., 2018; Matias Rodrigues et al., 2017; Quast et al., 2013). In total, $8.41 \times 10^5$ SSU and $1.09 \times 10^6$ LSU reads were obtained with averages of $9.34 \times 10^4$ SSU and $1.21 \times 10^5$ LSU reads per sample. Reads identified as chloroplast and mitochondrial ribosomal gene fragments were excluded from downstream analysis. Preliminary taxonomic analysis displayed unusually high numbers of SSU reads and inconsistencies between SSU and LSU



results for CY26. Analysis of read abundance revealed, that, in contrast to the other samples, the data of CY26 contained a

high number of reads starting with sequences of degenerated random 16S primers, which are usually used in fingerprinting techniques (Sakallah et al., 1995). Thus, all 16S rRNA reads from sample CY26 were excluded.

All downstream analyses were performed in R version 3.6.3 (R-Development-Core-Team, 2011). The base R functions were used for data processing and plotting. The SSU and LSU files were imported into R using the biomformat package (McMurdie and Paulson, 2019). Additionally, the InterPro matches provided by the MGnify pipeline (Mitchell et al., 2018)

were specifically analyzed for viral, allergenic, and bacterial ice nucleating proteins.

## 2.6 Freezing experiments and data analysis

For ice nucleation measurements, aqueous extracts of two pieces of each air filter sample were analyzed using the Twin-plate Ice Nucleation Assay (TINA) (Kunert et al., 2018). The extracts were prepared with pure water, which was obtained from a Barnstead™ GenPure™ xCAD Plus water purification system (Thermo Scientific, Braunschweig, Germany), autoclaved

at 121 °C for 20 min, and filtered through a sterile 0.1 μm pore diameter polyethersulfone (PES) vacuum filter unit (VWR International, Radnor, PA, USA). For extract preparation, 20 mL of the pure water were added to each filter piece in sterile 50 mL tubes (SPL life sciences Co. Ltd, South Korea). The tubes were vortexed at 2700 rpm for 15 min and another 15 min upside down (Vortex-Genie 2, Scientific Industries Inc., USA). Afterwards, the aqueous extracts were transferred into new sterile 50 mL tubes. For IN characterization, 1 mL aliquots of the aqueous extracts were treated as follows: (i) filtration through

a 5 μm pore diameter filter (Acrodisc, PES, Pall GmbH, Dreieich, Germany), (ii) filtration through a 0.1 μm pore diameter filter (Acrodisc), (iii) incubation at 98 °C for 1 h, (iv) filtration through a 5 μm pore diameter filter and incubation at 98 °C for 1 h, and (v) filtration through a 0.1 μm pore diameter filter and incubation at 98 °C for 1 h.

Of each untreated and treated aqueous extract, 96 aliquots of 3 μL were pipetted by a liquid handling station (epMotion ep5073, Eppendorf, Hamburg, Germany) into 384-well plates (Eppendorf). The plates were placed in the TINA instrument,

and, after 1 min equilibration at 0 °C, they were cooled to -28 °C with a continuous cooling rate of 1 °C min$^{-1}$. Pure water samples (autoclaved and 0.1 μm filtered) served as negative controls in the individual experiments. Additionally, aqueous extracts of two filter pieces of one filter blank, one mounting blank, and one start-up blank served as negative controls for the aerosol sampling process (section 2.1). As shown in Fig. B1, the blank samples showed freezing events starting around -20 °C with most droplet freezing around -25 °C. Thus, the observed temperature range for the aerosol samples was 0 to -20 °C.

The freezing temperature data of the individual measurements were binned into 0.2 °C intervals, which corresponds to the temperature uncertainty of the measurement (Kunert et al., 2018). For each aerosol sample, the data of the 96 droplets of the two filter pieces were combined to 192 droplets for further calculations. All calculations and plotting was performed with R version 3.6.3 (R-Development-Core-Team, 2011). The fraction of frozen droplets ($f_{ice}$), the cumulative number of IN ($N_v$) per liter of air, and corresponding errors were calculated as described in Vali (1971) and Kunert et al. (2018).





## 150 3 Results and Discussion

### 3.1 Overall community composition and community changes

The composition of the airborne community was determined by taxonomic assignment of ribosomal RNA gene SSU (small subunit) and LSU (large subunit) reads. In total, we detected 3000 operational taxonomic units (OTUs) for SSU and 4167 OTUs for LSU reads. Note, that the SSU OTUs do not include data from sample CY26 (see section 2.5). The obtained OTUs, 155 however, can be seen as lower estimates of species richness, as MGnify assigns OTUs based on a reference database to different taxonomic levels. This leads to numerous OTUs at higher taxonomic levels, which comprise reads from multiple species that cannot be identified to lower taxonomic levels. Thus, for downstream analysis and inter-sample comparison, read proportions were used.

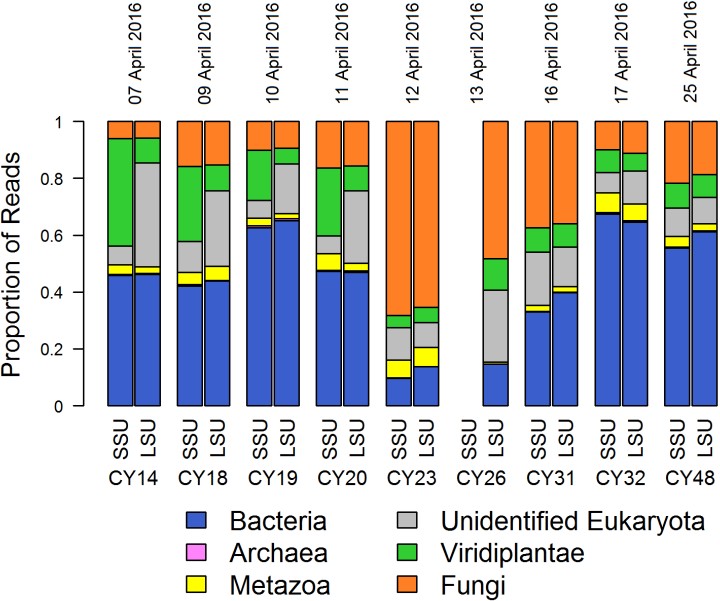

**Figure 1. Taxonomic composition.** Proportions of SSU and LSU reads assigned to different domains (Bacteria, Archaea, Unidentified Eukaryota) and kingdoms (Metazoa, Viridiplantae, Fungi) for all aerosol samples. SSU reads of CY26 were excluded because of a contamination by 16S rRNA gene amplicon sequences. Eukaryota reads, which were not assigned to a kingdom or not further classified, were grouped in "Unidentified Eukaryota". The proportion of Archaea reads was < 1 % for all samples.

Figure 1 shows the read proportions and taxonomic assignments on domain and kingdom level for the individual samples. 160 The taxonomic composition, in particular the read proportion of fungi and bacteria, changed with sample CY23, indicating a change of the airborne community. Sample CY23 was collected after a 3 h rain event, which corresponds to the collection end of CY20 (Fig. A1). Moreover, two additional rain events occurred during sampling of CY23. This sample shows the





highest proportion of fungal reads (> 60 %) and lowest read proportion of Bacteria (< 20 %) and Viridiplantae (< 10 %). Samples collected up to four days after the rain events (CY26 and CY31) also show higher proportions of fungal reads (up to 50 %)

and lower read proportions for Bacteria and Viridiplantae than the other samples. Metazoa and Archaea read proportions were < 7 % and < 1 %, respectively, in all samples (Fig. 1) .

The high proportion of fungal reads in CY23 may result from increased concentrations of fungal spores following rainfall, as many fungal spores are dispersed by rain splash, rain tap and puff, or by other spore discharge mechanisms that require water (e.g., Hirst, 1953; Hirst and Stedman, 1963; Gregory et al., 1959; Lacey, 1996; Jones and Harrison, 2004; Crandall and Gilbert,

2017; Kauserud et al., 2005; Kaygusuz and Faruk Çolak, 2017). Bacteria are often associated with larger particles such as soil dust and plant fragments or form aggregates with each other (Shaffer and Lighthart, 1997; Polymenakou et al., 2008; Reche et al., 2018). As larger particles are more efficiently removed by precipitation than smaller particles, because the latter tend to follow the air stream around the raindrops (Pranesha and Kamra, 1997; Seinfeld and Pandis, 2016), an efficient washout is assumed for bacteria and larger particles. This facilitates the return of airborne bacteria and other biological particles to

the Earth´s surface as well as deposition in new habitats (Peter et al., 2014; Reche et al., 2018), which possibly explains the reduced read proportions for Bacteria, Viridiplantae, and Unidentified Eukaryota in samples CY23 and CY26.

Figure 1 also shows for the individual samples, that the proportions of SSU and LSU reads assigned to Archaea, Bacteria, Fungi, and Metazoa are comparable, indicating that both genes can be used as representative markers for these groups. In contrast, Viridiplantae and Unidentified Eukaryota show differences in the SSU and LSU read proportions in samples CY14-

CY20 and more equal read proportions in all other samples. This indicates a change of the airborne community due to the rain events and a lack of plant LSU sequences in the SILVA reference database.

Unlike Pro- and Eukaryotes, for which ribosomal RNA genes can be used for identification, viruses do not contain common genetic markers (Edwards and Rohwer, 2005). Analysis of the InterPro matches, however, revealed the presence of various double and single stranded DNA and RNA viruses in the aerosol samples. Of the virus-related InterPro matches ∼ 44 % were

assigned to viruses, which infect bacteria and archaea (e.g., Siphoviridae, Podoviridae, Myoviridae, and Microviridae) and cover ∼ 80 % of the reads that were assigned to virus-related matches. Viruses infecting animals and humans (e.g., Adenoviridae, Baculoviridae, Flaviviridae, Herpesviridae, Poxviridae) exhibited also ∼ 44 % of the matches corresponding to ∼ 4 % of reads. Plant viruses such as Geminiviridae, Caulimoviridae, and Potyviridae were less abundant with ∼ 7 % of the matches and < 1 % of reads. The remaining ∼ 4 % of the virus-related InterPro matches (∼ 15 % reads) could either not be assigned to hosts

or were from fungi and amoeba infecting viruses. The detection of RNA viruses (e.g., Flaviviridae, Reoviridae, Retroviridae) can be explained by the integration of these viruses into the genomes of their host cells and their persistence as proviruses (Edwards and Rohwer, 2005).

### 3.1.1  Composition and community changes of Fungi, Viridiplantae, and Metazoa

The composition and community changes of Fungi, Viridiplantae, and Metazoa were further analyzed on class level. Figure 2

shows, that most fungal reads could be attributed to different classes and phyla of Ascomycota and Basidiomycota. Large fractions of Ascomycota reads (up to 40 %) could not be attributed to any Ascomycota class. All samples show higher read





**Figure 2. Taxonomic composition of Fungi.** Taxonomic classification and proportion of SSU and LSU reads of Fungi at class level.

proportions for Ascomycota than for Basidiomycota. Sample CY23, however, shows with around 38 % for SSU and 32 % for LSU, the highest Basidiomycota read proportions of all samples. Of all Basidiomycota reads of sample CY23, ∼ 79 % (∼ 30 % of fungal reads) were assigned to the class Agaricomycetes. The Agaricomycetes read proportion is also enhanced in sample

CY26 (∼ 73 % of Basidiomycota; ∼ 8.5 % of Fungi). The Agaricomycetes are wood-degrading fungi, and many of them form fruiting bodies, which release spores in response to humidity changes (Zoberi, 1964; Elbert et al., 2007; Hassett et al., 2015; Löbs et al., 2020). Analysis of the InterPro matches revealed higher read counts for the allergen Alt 1 from *Alternaria alternata* (Dothideomycetes) for samples collected during and after the rainfall period (Table C1). *Alternaria* is a typical airborne fungus, for which the onset of rain can lead to a period of increased spore concentrations through a tap and puff mechanism (Hirst and

Stedman, 1963; Lacey, 1996).

For the Viridiplantae (green plants), most reads (up to 92 %) could not be classified on class level (Fig. 3) indicating a lack of reference sequences in the SILVA database. All samples show large proportions of reads assigned to the Streptophyta, which



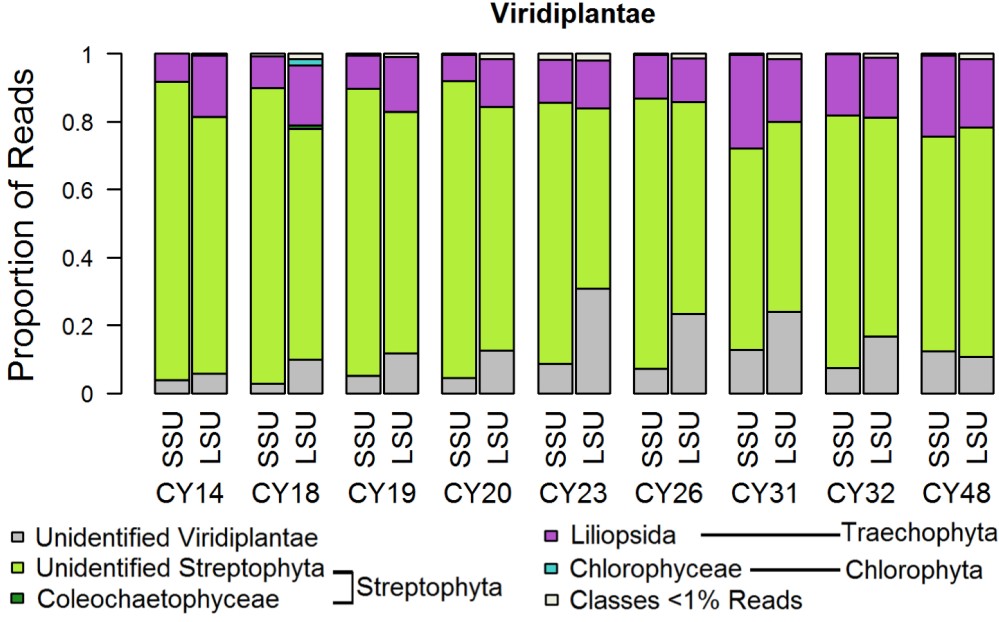

**Figure 3. Taxonomic composition of Viridiplantae.** Taxonomic classification and proportion of SSU and LSU reads of Viridiplantae at class level.

include land plants and green algae. Between $\sim 8\,\%$ and $\sim 28\,\%$ of the sample reads were assigned to the class Liliopsida of the Tracheophyta (vascular plants), which comprises lilies, grasses, palms, orchids, and other monocotyledon plants. Samples

CY23, CY26, CY31, and CY32 show higher differences in SSU and LSU proportions for Unidentified Viridiplantae than the other samples. Although the limited taxonomic classification does not allow specific discussion, this change of proportions suggests a community change of plant-derived aerosol particles, which can be pollen, moss and fern spores, achene fibers, bark, seeds, algae or fragments like leaf tissues. Pollen, however, are suggested to be the most abundant plant bioaerosols in the atmosphere (Rittenour et al., 2012). The effects of precipitation on pollen concentrations depend primarily on rainfall intensity.

While decreased pollen concentrations are associated with more intensive rain events of at least $5\,mm\,h^{-1}$ (Ribeiro et al., 2003; Rathnayake et al., 2017; Kluska et al., 2020), increased pollen concentrations have been measured immediately before and during rainfall (Puls and Von Wahl, 1991; Kluska et al., 2020). The pollen emission at the onset of rain can be caused by rain drops hitting the stamina of the flowers, which are ready for pollen emission (Puls and Von Wahl, 1991).

     As various airborne pollen species are important allergens, the InterPro matches were analyzed for allergenic protein

encoding-reads. As detailed in Table C1, twelve possible allergen types were found; all from plants that are common in Cyprus. The most abundant allergens were Amb allergen type, Lol p 1, Bet v 1 type, and Par j 1/2. The Amb allergen type was found in all samples. Amb a-type allergens are major allergens in pollen of ragweed (*Ambrosia artemisiifolia*, Asteraceae) and Cupressaceae trees (e.g., Mediterranean cypress, mountain cedar), which are abundant in Cyprus (Wolf et al., 2017; Gucel et al., 2013).





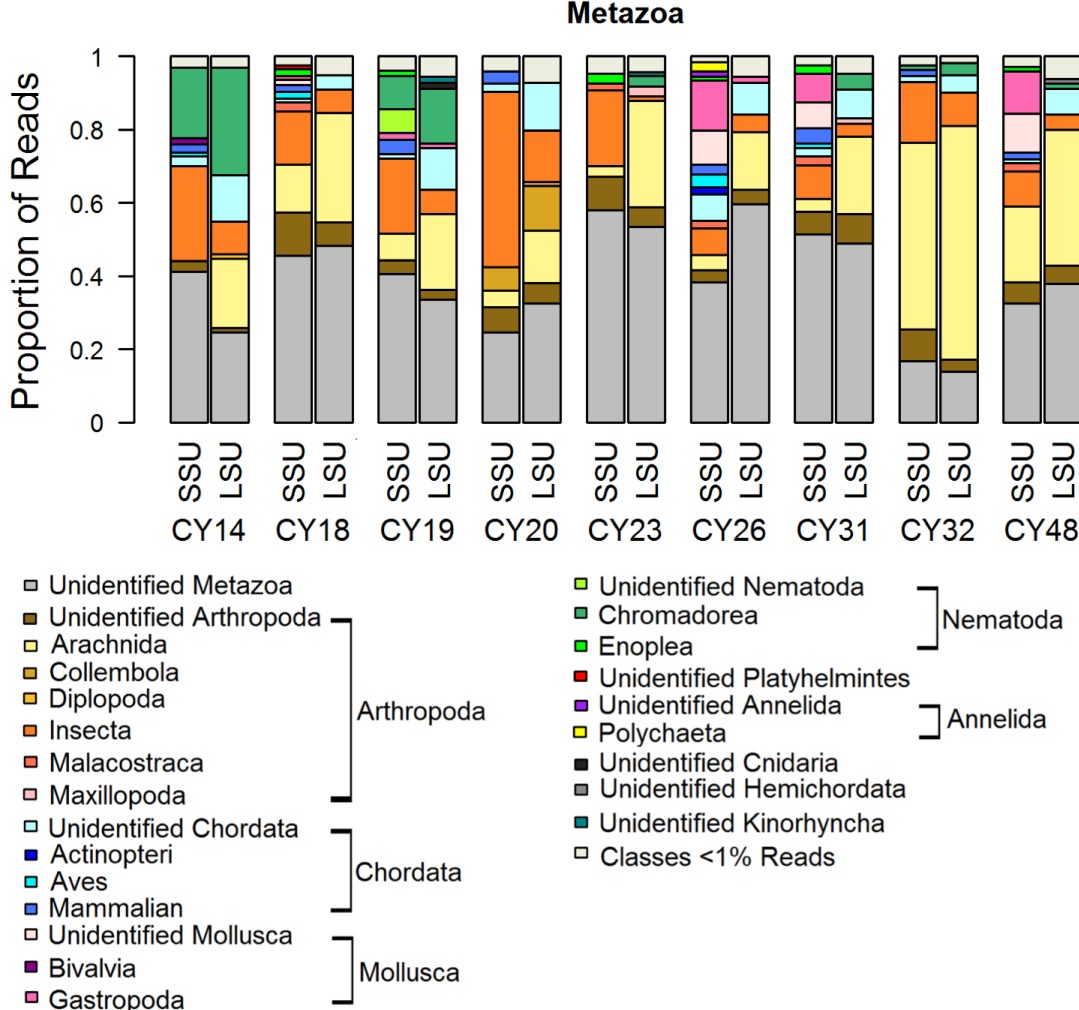

**Figure 4. Taxonomic composition of Metazoa.** Taxonomic classification and proportion of SSU and LSU reads of Metazoa at class level.

The allergen Lol p 1 was found in all samples, except CY19. Lol p 1 is the major allergen of the perennial ryegrass (*Lolium perenne*, Poaceae), which is common in Cyprus and flowers in spring time (Gucel et al., 2013; Herridge et al., 2021), which was the period of sampling. The Bet v 1-type allergens can be found in members of various plant families such as Betulaceae, Fabaceae, and Rosaceae, which are all common in Cyprus. Samples CY26 and CY48 had with 63 and 20, respectively, higher read counts for Bet v 1-type allergens than all other samples (0-9 reads). The allergens Par j 1/2 from *Parietaria judaica* exhibited the highest read counts (154) in CY31. *Parietaria judaica*, a weed of the Urticaceae family and found throughout Cyprus, has a flowering time in spring and autumn and is one of the most important allergens in the Mediterranean area (Colombo et al., 2003; Gucel et al., 2013; Ozturk et al., 2013; Ciprandi et al., 2018). The detected plant allergens allow an additional insight





into the origin of the airborne plant particles to the taxonomic analysis of ribosomal reads, but show no clear trends with regard to rain.

Metazoa show a diverse picture on class level with high proportions of up to 60 % of unidentified reads as well as differences in the proportions of SSU and LSU reads (Fig. 4). The identified metazoan classes comprised microbial metazoa (e.g., nematodes (Bik, 2019)) and larger organisms (e.g., insects, spiders, snails). The analysis of the InterPro matches revealed reads encoding for allergens mainly from wasps (Ves m 5/Ves v 5/Ves 5 type) and bees (Api m 3) but also from mites (e.g., Der f 7), nematodes (ABA 1), and mammals (e.g., Fel d 1) as listed in Table C1. All samples show a high proportion of reads assigned to Insecta and Arachnida (includes spiders and mites) and matches to their allergens (Fig. 4, Table C1). Two samples (CY14 and CY19) additionally show larger proportions of nematode reads and matches for nematode allergens. Nematodes are abundant in soil and aquatic environments or can be parasites in plants or animals (Seesao et al., 2017). They can be readily dispersed by the wind as living organisms or resting states, which they produce to survive unfavorable conditions such as dehydration (Nkem et al., 2006; Vanschoenwinkel et al., 2008; Carroll and Viglierchio, 1981; Ptatscheck et al., 2018). The higher read proportions for Gastropoda (snails and slugs) in samples CY26, CY31, and CY48 might be explained by the presence of tissue or skin fragments in the air, due to their dispersal or transport by birds during the nesting phase (Pearce et al., 2010; Shikov and Vinogradov, 2013).

### 3.1.2 Composition and community changes of Bacteria and Archaea

Figure 5 shows the taxonomic composition of Bacteria on class level, which is a typical composition of bacterial communities in air (e.g., Tong and Lighthart, 2000; Bowers et al., 2011; Jeon et al., 2011; Bowers et al., 2012; Robertson et al., 2013; Smith et al., 2013; Yooseph et al., 2013; Jang et al., 2018; Mescioglu et al., 2019). Large proportions of reads were assigned to the Actinobacteria and Firmicutes. At a closer look, the Firmicutes were the dominating prokaryotic phylum, except in sample CY23 and CY26. Taxa within the Firmicutes might be linked to anthropogenic influences, as they are commonly found in soil and in the human gut microbiome (Kho and Lal, 2018; Almeida et al., 2019).

Interestingly, for samples CY23 and CY26, the proportion of Unidentified Bacteria reads is with 40 to 50 % more than twice as high as for all other samples (< 20 %). This might indicate that, for example, soil bacteria were emitted and aerosolized by the impact of raindrops on soil (Joung et al., 2017). The emitted bacterial taxa, however, were not represented in the reference database and remained unidentified. Overall, the results are consistent with the findings of Zhen et al. (2017), who reported temporary shifts of the airborne bacterial community depending on rain intensity, but did not further specify the community structure.

Figure 6 shows that most archaea reads were assigned to the Euryarchaeota, and a small proportion was assigned to the Thaumarchaeota. Similar to bacteria, high proportions of up to 50 % of unidentified reads for samples CY23 and CY26 might indicate some emission of archaea by the impact of rain drops on soil or water surfaces. Within the Euryarchaeota, Halobacteria reads dominated. Halobacteria belong to the extreme halophilic Archaea, which can be found in hypersaline environments (> 10 % salt up to saturation) such as salt lakes or saline sediments (Grant and Ross, 1986; Fendrihan et al., 2007; Oren, 2014). The various salt lakes (e.g., Larnaka salt lake) in Cyprus might be a source of the detected airborne Halobacteria. Moreover,





**Figure 5. Taxonomic composition of Bacteria.** Taxonomic classification and proportion of SSU and LSU reads of Bacteria at class level. SSU reads of CY26 were excluded because of a contamination by 16S rRNA gene amplicon sequences.

the detection of methanogens indicates possible anthropogenic influences, as methanogens are not only found in terrestrial and aquatic environments but also in the intestines of larger organisms (Liu and Whitman, 2008; Söllinger and Urich, 2019). Thus, they might stem from the animal waste treatment company in Cyprus, which is close to the station, or from waste water possibly at sewage treatment plants, or marine pollution by hotels or boats. Moreover, airborne Methanobacteria have been

found during fertilisation periods (Fröhlich-Nowoisky et al., 2014; Wehking et al., 2018).





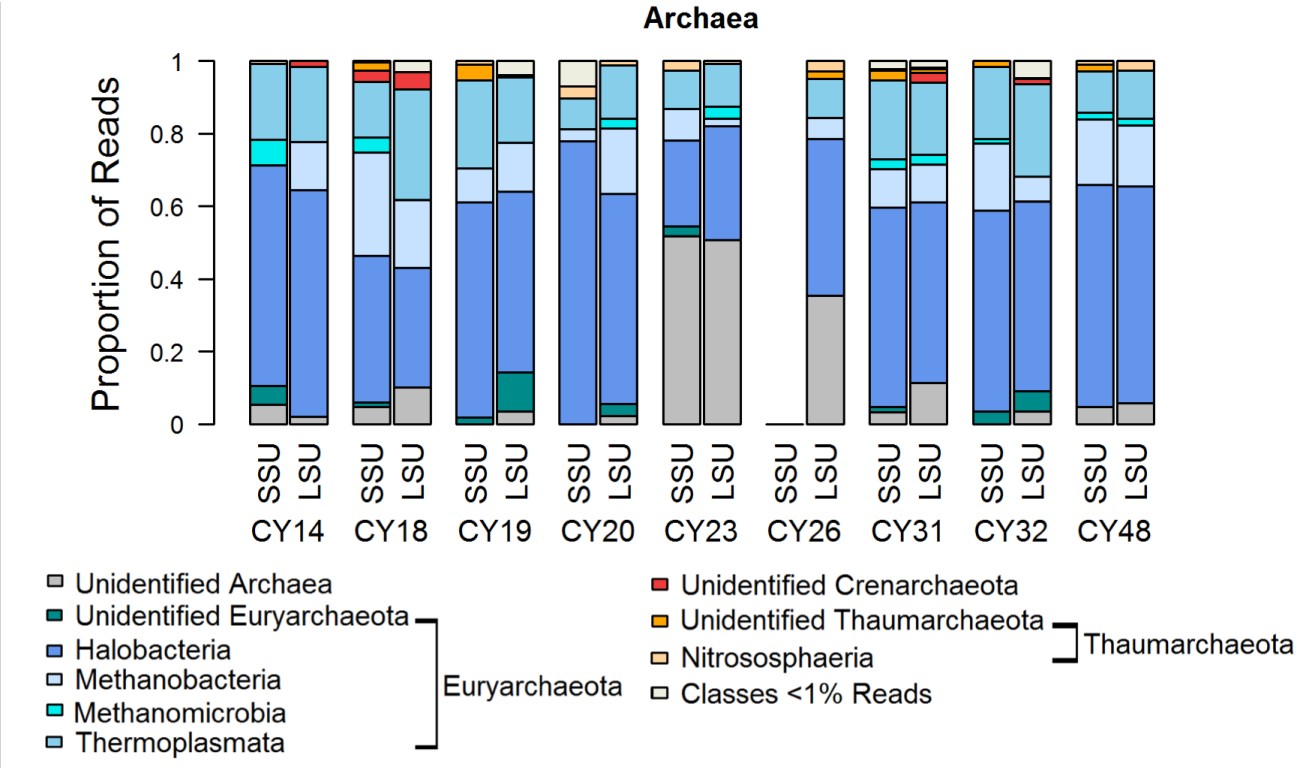

**Figure 6. Taxonomic composition of Archaea.** Taxonomic classification and proportion of SSU and LSU reads of Archaea at class level.

## 3.2 Ice nucleation activity

Figure 7 provides an overview of the freezing abilities for the individual samples. Figure 7a shows the fraction of frozen droplets, Fig. 7b shows the median temperatures $T_{50}$ (i.e., the temperature at which 50 % of the droplets were frozen), and Fig. 7c shows the ice nuclei (IN) concentrations calculated per liter volume of sampled air (from the data shown in Fig. 7a).

The individual samples initiated freezing between -6.3 °C and -11.3 °C. The frozen fraction curves show differences in shapes and slopes as well as different $T_{50}$ values. The curve of sample CY23, which was collected after and during the rain events, approximates a straight line with $T_{50}$ = -16.7 °C. Contrastingly, the samples CY26-CY32, which were collected after the rain events, show all a similar convex curve shape and higher median freezing temperatures ($T_{50}$ = -13.9 °C to -15.5 °C). All other samples, including sample CY48, which was collected 13 days after the last rain event, show concave curves and lower $T_{50}$

values (-17.3 °C to -18.3 °C).

Figure 7c shows that the individual samples differ in their IN concentrations at given temperatures. For example, at -15 °C, between $10^{-1}$ and $10^{-2}$ IN per liter of sampled air were found for the investigated samples. Overall, the obtained IN concentrations span about two orders of magnitude for temperatures > -15 °C and about one order of magnitude for temperatures





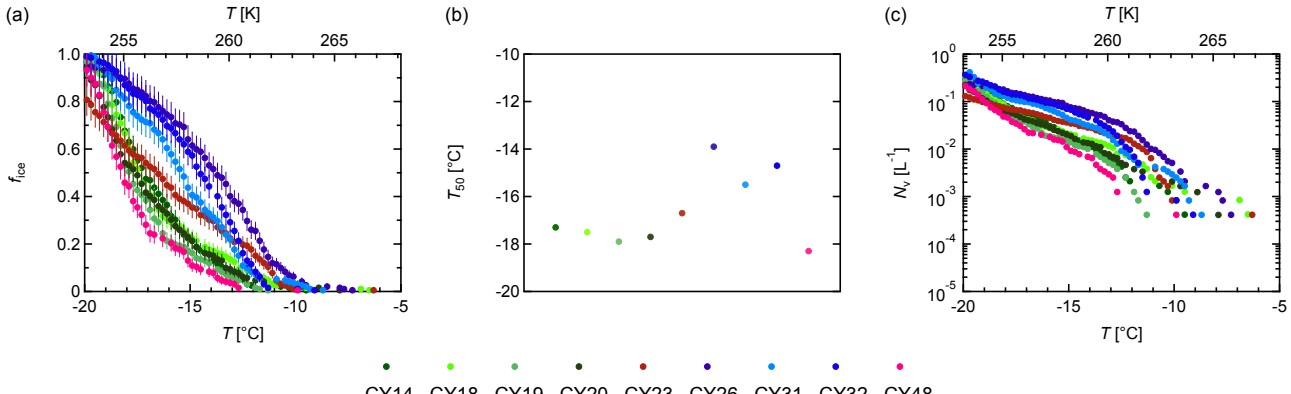

**Figure 7. Ice nucleation activity.** Fraction of frozen droplets ($f_{ice}$) (a) and cumulative number of IN ($N_v$) per liter of air (b) vs. temperature ($T$) for all aerosol samples. The colors represent individual samples; the error bars were calculated using the counting error and the Gaussian error propagation.

$< -15\,°C$. These results are in good agreement with Gong et al. (2019), who reported IN concentrations in the same order of

magnitude for samples collected in Cyprus in April 2017.

The results indicate heterogeneity of the IN compositions in the samples. Higher initial freezing temperatures and $T_{50}$ values can be an indication for the presence of bio-IN, as many bio-IN are known to nucleate at temperatures $> -15\,°C$ while mineral IN dominate at $< -15\,°C$ (e.g., Maki et al., 1974; Kieft and Ahmadjian, 1989; DeMott and Prenni, 2010; Atkinson et al., 2013; Hiranuma et al., 2013; Joly et al., 2013; Pummer et al., 2015; Fröhlich-Nowoisky et al., 2016; Kunert et al., 2019; Huang et al.,

290   2021).

To further characterize the IN of the different aerosol samples, filtration and heat treatment experiments were performed with the aqueous extracts. Figure 8 shows that filtration through a 5 µm and a 0.1 µm filter reduced the ice nucleation activity in all samples, indicating that all samples contained IN larger than 5 µm, IN between 0.1 µm and 5 µm, and IN smaller than 0.1 µm. For samples CY14, CY23, CY26, CY31, and CY48, however, the results show only a small fraction of IN between 0.1 µm and

5 µm, while many IN were smaller than 0.1 µm. Of all samples, CY18 and CY19 show the lowest percentage of frozen droplets after 0.1 µm filtration, and thus, the lowest numbers of IN smaller than 0.1 µm in the observed temperature range.

Heat sensitivity can indicate biological IN, as heat can reduce ice nucleation activity of many bio-IN (Pouleur et al., 1992; Henderson-Begg et al., 2009; Morris et al., 2013; Fröhlich-Nowoisky et al., 2015; Pummer et al., 2015; Hara et al., 2016a; Kunert et al., 2019). Certain bio-IN, however, such as Lysinibacillus, a heat-resistant isolate of *Xanthomonas*, and various

plant pollen, as well as inorganic particles are heat resistant to temperatures $> 90\,°C$ (Pummer et al., 2012; Hill et al., 2014; Failor et al., 2017). Heat treatment at $98\,°C$ for 1 h reduced the ice nucleation activity of the individual samples to a different extent (Fig. 8). For example, the ice nucleation activity of samples CY20-CY32 was significantly reduced in the different size fractions after heat treatment (Fig. 8d-h), indicating the presence of heat-sensitive IN and thus presumably bio-IN of different sizes. Many fungal spores and bacterial aggregates are larger than 5 µm, whereas single bacterial cells or cell fragments can







**Figure 8. Effects of filtration and heat treatment on ice nucleation activity.** Fraction of frozen droplets ($f_{ice}$) vs. temperature ($T$) for all aerosol samples (a-i). The color represents individual treatments; the error bars were calculated using the counting error and the Gaussian error propagation.

be smaller than 5 μm, and ice nucleating macromolecules are typically in the submicrometer range (Fröhlich-Nowoisky et al., 2016; Pummer et al., 2012, 2015; Šantl-Temkiv et al., 2015; Felgitsch et al., 2018). The samples CY23 and CY26, which were collected during and after the rain events, appear to have the largest fraction of heat-sensitive IN larger than 5 μm. In contrast,





heat did only slightly affect samples CY18 and CY19 (Fig. 8b-c), indicating a significant fraction of heat-resistant IN in these samples, possibly due to particles from the dust events, which were registered during the collection of these samples (Fig. A1).

For samples CY14 and CY48, the ice nucleation activity of 5 µm and 0.1 µm filtrates was reduced by heat, while the unfiltered extract was only slightly affected (Fig. 8a,i). The results suggest, that in these samples, many heat-resistant IN were larger than 5 µm, whereas many heat-sensitive IN were smaller than 0.1 µm, or, for CY14, also between 0.1 µm and 5 µm.

### 3.3 Possible sources of biological ice nuclei

The results of the freezing experiments indicate the presence of bio-IN in different size ranges. These bio-IN may be present

as living or dead cells, fungal hyphae and spores, pollen, cell fragments, detached macromolecules, or associated with plant particles or soil organic matter (Schnell and Vali, 1976; Pummer et al., 2012; Fröhlich-Nowoisky et al., 2015; Šantl-Temkiv et al., 2015; O'Sullivan et al., 2016; Hill et al., 2016; Conen and Yakutin, 2018). Note that not all of these IN contain DNA, as for example ice nucleating macromolecules or cell membrane fragments with attached IN proteins, and are thus not covered by DNA analysis. The largest fraction of bio-IN > 5 µm were detected in CY23 and CY26. For these samples, the sequencing

results show a higher proportion of reads assigned to fungi, in particular Agaricomycetes (Fig. 1, 2), suggesting a possible contribution of these fungi to the increase of IN. Systematic surveys suggest that fungal ice nucleation activity, in particular at temperatures > -15 °C, is restricted to a limited number of fungal species (Fröhlich-Nowoisky et al., 2015; Haga et al., 2013, 2014; Iannone et al., 2011; Jayaweera and Flanagan, 1982; Kieft and Ahmadjian, 1989; Kunert et al., 2019; Morris et al., 2013; Pouleur et al., 1992; Pummer et al., 2013, 2015). The class Agaricomycetes contains approximately 21 000 fungal

species (Hibbett et al., 2014), but only a handful has been tested for ice nucleation activity so far (Pummer et al., 2013; Haga et al., 2014). Further surveys of ice nucleation properties of Agaricomycetes and other fungi are necessary.

To identify possible bio-IN sources, the SSU and LSU data were analyzed for known IN fungi and bacteria that are active at temperatures > -15 °C. In total, 4 fungal genera and 13 bacterial genera that contain IN-active species were detected, but all exhibited read proportions < 1 % and species identification was partly not possible. Note that not all species and strains of the

identified genera are IN-active, and, especially in bacteria, not all cells express the IN gene (Lindow et al., 1982; Kunert et al., 2019). Moreover, some recently isolated IN-active fungi and bacteria such as *Cryptococcus* sp. and *Brevibacterium* sp. are not identified to species level (Beall et al., 2021). Thus, the following microorganisms represent possible sources of IN but provide no proof to what extent they were IN-active in the air.

Among the fungi were *Cryptococcus* sp., *Fusarium* sp., the soil-fungus *Mortierella alpina*, and five species of the plant-

pathogen rust fungi *Puccinia* (*P. arachidis, P. hordei, P. horiana, P. striiformis, P. triticina*). Spores of *Puccinia striiformis* (stripe rust or yellow rust) and *P. triticina* can have initial freezing temperatures of -7 °C and -9 °C, respectively (Morris et al., 2013). Although several other rust species are known as ice nucleators, for *P. arachidis* (peanut rust), *P. hordei* (barley leaf rust), and *P. horiana* (chrysanthemum white rust), ice nucleation activity has not yet been reported. In contrast to *Puccinia* spp., for which the spores are IN-active, the IN of *M. alpina* and *Fusarium* spp. are cell-free proteinous macromolecules and can trigger

freezing at temperatures as high as -5 °C or -1 °C, respectively (Pouleur et al., 1992; Richard et al., 1996; Fröhlich-Nowoisky





et al., 2015; Kunert et al., 2019). The macromolecules might be released into the surrounding environment and contribute to atmospheric IN.

Among the potential IN-active bacteria were the well-known genera and species *Erwinia*, *Pantoea agglomerans*, *Pseudomonas*, and *Xanthomonas* as well as the recently identified *Brevibacterium*, *Idiomarina*, *Lysinibacillus*, *Planococcus maritimus*, *Psychrobacter* (Maki et al., 1974; Govindarajan and Lindow, 1988; Hill et al., 2014; Failor et al., 2017; Beall et al., 2021). The genus *Pseudomonas*, which includes a range of IN-active species, was detected in all samples. Most of the reads (∼ 94 %) assigned to the genus *Pseudomonas*, however, could not be identified to species level. The remaining 58 reads were distributed over 12 species (1 to 14 reads per species), but did not include *P. syringae* a common bacterial ice nucleator. Analysis of the InterPro matches for IN-active proteins, however, revealed one InterPro match (IPR000258, bacterial ice-nucleation, octamer repeat) for in total 24 reads from six of the nine samples (1 to 12 reads per sample) indicating the presence of a bacterial IN-active species. Recently, ice nucleation activity was found in *Halococcus morrhuae* and *Haloferax sulfurifontis*; two species of Haloarchaea, which initiated the freezing of water droplets at temperatures as high as –18 °C (Creamean et al., 2021). In our data set, however, only one read from CY48 was assigned to the genus Halococcus, while Haloferax was not detected.

## 4 Conclusions

Our results show that rainfall in a rural dryland can lead to short-term changes of bioaerosol and IN composition. Particle washout and fungal spore discharge in response to humidity changes appear to play a major role for the observed composition changes, in particular for fungi and bacteria. The results of the freezing experiments suggest a rain-related enhancement of bio-IN > 5 µm and < 0.1 µm. Microorganisms that are known to be IN-active, however, could only rarely been detected. This may reflect unknown sources of atmospheric bio-IN and the presence of cell fragments or cell-free IN macromolecules that do not contain DNA, as known for many IN-active fungi. Similar observations of increased bioaerosol, fungal spore, and IN concentrations in relation to rainfall have been reported for different ecosystems indicating a strong coupling of rainfall and particle emission (Huffman et al., 2013; Prenni et al., 2013; Schumacher et al., 2013; Tobo et al., 2013; Wright et al., 2014; Bigg et al., 2015; Heo et al., 2014; Gosselin et al., 2016; Rathnayake et al., 2017). The release of bioaerosols during and after rain may play an important role in the spread and reproduction of microorganisms and may contribute to the atmospheric transmission of pathogenic and allergenic particles. In view of the relevance of bioparticles as pathogens and allergens as well as nuclei for clouds and precipitation (bioprecipitation cycle) (Bigg et al., 2015; Morris et al., 2014, 2017) and potential future developments in the Anthropocene (Crutzen and Stoermer, 2000; Pöschl and Shiraiwa, 2015; Rodriguez-Caballero et al., 2018), further investigations are required to understand the sources, properties, and short-term dynamics of bioaerosols and bio-IN in various ecosystems. In particular, further investigations are necessary for the identification of the IN macromolecules and for the detection and quantification of these IN in soil, precipitation, and atmospheric samples.





*Data availability.* The metagenomic data have been deposited at https://www.ebi.ac.uk/metagenomics/ under: MGYA00166510 (CY14), MGYA00199054 (CY18), MGYA00166484 (CY19), MGYA00166504 (CY20), MGYA00166527 (CY23), MGYA00166526 (CY26), MGYA00166493 (CY31), MGYA00199055 (CY32), MGYA00166518 (CY48). The IN data are available at Ed-

mond – the Open Access Data Repository of the Max Planck Society, under https://dx.doi.org/10.17617/3.6r.





## Appendix A: Aerosol samples and meteorological conditions.

**Table A1. Overview of air filter samples.** Sample names, sampling periods, and sampled air volumes at standard conditions (0 °C, 1013 hPa).

| Sample | Start of Sampling | | End of Sampling | | Sampling Time | Sampled Air Volume |
|---|---|---|---|---|---|---|
| | Date | Time [LT] | Date | Time [LT] | [min] | [m$^3$] |
| CY14 | 07 April 2016 | 09:41 | 08 April 2016 | 09:41 | 1440 | 1338 |
| CY18 | 09 April 2016 | 09:52 | 10 April 2016 | 09:52 | 1440 | 1330 |
| CY19 | 10 April 2016 | 09:54 | 11 April 2016 | 09:52 | 1440 | 1350 |
| CY20 | 11 April 2016 | 09:54 | 12 April 2016 | 09:52 | 1440 | 1358 |
| CY23 | 12 April 2016 | 10:02 | 13 April 2016 | 09:56 | 1429 | 1350 |
| CY26 | 13 April 2016 | 10:01 | 14 April 2016 | 10:01 | 1440 | 1350 |
| CY31 | 16 April 2016 | 10:07 | 17 April 2016 | 10:07 | 1440 | 1351 |
| CY32 | 17 April 2016 | 10:07 | 18 April 2016 | 10:07 | 1440 | 1344 |
| CY48 | 25 April 2016 | 10:01 | 26 April 2016 | 10:01 | 1440 | 1349 |



**Figure A1. Meteorological conditions and aerosol mass concentrations.** One-hour averages of temperature (T), relative humidity (RH), rain fall (Rain), wind speed (WS), and aerosol mass concentrations of $PM_{10}$ and $PM_{2.5}$ during the sampling campaign.





**Appendix B: Freezing experiments with pure water and sampling blanks.**

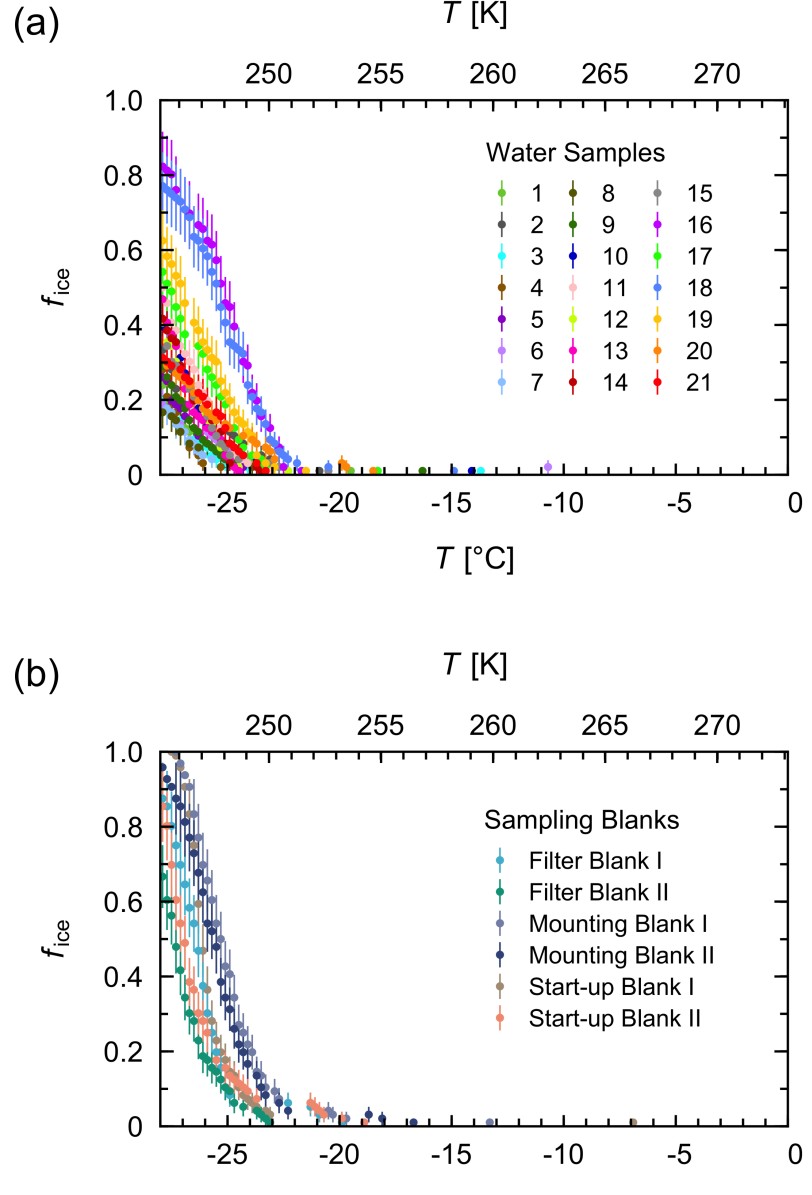

**Figure B1. Negative controls of the freezing experiments.** Fraction of frozen droplets ($f_{ice}$) vs. temperature ($T$) for all water samples (a) and different types of sampling blanks that served as negative controls (b). For each type of sampling blank, two filter pieces were measured. The error bars were calculated using the counting error and the Gaussian error propagation.





## Appendix C: InterPro matches and read counts for allergenic proteins.

**Table C1. Protein-encoding reads for allergens.** Overview of InterPro matches and read counts for allergenic protein-encoding reads. IPR indicates the InterPro accession number.

| IPR | Allergens | Species/Family | CY14 | CY18 | CY19 | CY20 | CY23 | CY26 | CY31 | CY32 | CY48 |
|---|---|---|---|---|---|---|---|---|---|---|---|
| | | Fungi | | | | | | | | | |
| IPR032382 | Alt 1 | *Alternaria alternata* | 0 | 8 | 0 | 0 | 15 | 23 | 23 | 16 | 7 |
| | | Viridiplantae | | | | | | | | | |
| IPR005611 | Amb 5 | *Ambrosia* sp. | 4 | 9 | 0 | 0 | 3 | 4 | 3 | 0 | 1 |
| IPR018082 | Amb | Asteraceae, Cupressaceae | 30 | 56 | 31 | 31 | 22 | 86 | 31 | 11 | 35 |
| IPR024949 | Bet v1 type | Betulaceae, Apiaceae, Rosaceae, Asparagaceae, Fabaceae, Solanaceae | 0 | 9 | 4 | 9 | 8 | 63 | 7 | 7 | 20 |
| IPR005795 | Lol p1 | *Lolium perenne* | 5 | 17 | 0 | 24 | 2 | 24 | 12 | 10 | 11 |
| IPR005453 | Lol p2 | *Lolium perenne* | 0 | 1 | 0 | 0 | 0 | 0 | 1 | 0 | 0 |
| IPR006040 | Ole e1 | *Olea europaea* | 0 | 0 | 0 | 0 | 0 | 0 | 3 | 0 | 4 |
| IPR035506 | Pollen aller-gen/Os | *Oryza sativa* | 1 | 4 | 1 | 2 | 3 | 2 | 9 | 0 | 17 |
| IPR000528 | Par j1/2 | *Parietaria judaica* | 11 | 2 | 5 | 0 | 14 | 20 | 154 | 0 | 18 |
| IPR002411 | Amyl Inhib | Poaceae | 0 | 0 | 0 | 0 | 0 | 2 | 0 | 0 | 0 |
| IPR002914 | Lol p5, Pha a5, Poa p9, Phl p5b, Phl p6 | Poaceae | 0 | 3 | 0 | 0 | 0 | 4 | 0 | 0 | 8 |
| IPR006106 | Soft/Tryp Amyl Inhib CS | Poaceae | 0 | 14 | 0 | 0 | 0 | 1 | 4 | 0 | 4 |
| IPR006105 | Tryp Amyl In-hib CS | Poaceae | 0 | 15 | 0 | 0 | 1 | 0 | 2 | 0 | 0 |
| | | Metazoa | | | | | | | | | |
| IPR032487 | ABA 1, DvA1 | Ascarididae, Dictyocaulidae | 4 | 0 | 2 | 0 | 0 | 0 | 0 | 0 | 0 |

*continued on next page*





*continued from previous page*

| IPR | Allergens | Species/Family | CY14 | CY18 | CY19 | CY20 | CY23 | CY26 | CY31 | CY32 | CY48 |
|---|---|---|---|---|---|---|---|---|---|---|---|
| IPR002116 | Api m3 | *Apis mellifera* | 0 | 0 | 19 | 2 | 0 | 0 | 0 | 21 | 1 |
| IPR002450 | Bos d2, Can f2 | Bovidae, Canidae | 0 | 0 | 0 | 0 | 0 | 0 | 1 | 0 | 0 |
| IPR020234 | Der f7 | *Dermatophago-ides farinae* | 10 | 1 | 1 | 0 | 2 | 1 | 0 | 0 | 12 |
| IPR020306 | Der p5, Blo t5, Blo t21 | *Dermatophago-ides pteronyssi-nus, Blomia tropicalis* | 0 | 17 | 0 | 3 | 0 | 6 | 0 | 8 | 6 |
| IPR006178 | Fel d1 (chain 1) | *Felis silvestris* | 0 | 0 | 0 | 0 | 0 | 0 | 1 | 0 | 0 |
| IPR015332 | Fel d1 (chain 2) | *Felis silvestris* | 0 | 0 | 0 | 0 | 0 | 1 | 0 | 0 | 0 |
| IPR010629 | NSP | *Pieris rapae* | 0 | 2 | 0 | 6 | 0 | 0 | 2 | 0 | 0 |
| IPR018244 | Ves m5 /Ves v5 (CS) | Vespidae | 34 | 149 | 138 | 72 | 221 | 188 | 387 | 80 | 193 |
| IPR001283 | Ves m5 /Ves v5 | Vespidae | 48 | 143 | 59 | 48 | 159 | 175 | 377 | 60 | 192 |
| IPR002413 | Ves 5 type | Vespidae, Formicidae | 5 | 21 | 9 | 2 | 27 | 48 | 91 | 2 | 44 |



*Author contributions.* JF-N, UP, and BW designed the study. PY performed the sample collection and DNA extraction. KT performed the freezing experiments. BS-P, DP, JW, JF-N, KT, and PY analyzed the taxonomic data. BS-P and JF-N analyzed the InterPro matches. KT and

ATB analyzed the IN data. All authors discussed the results. JF-N wrote the manuscript with contributions from all co-authors.

*Competing interests.* The authors declare that they have no conflict of interest.

*Acknowledgements.* We thank the entire INUIT-BACCHUS-ACTRIS campaign team for their cooperation and support, the Cyprus Department of Labour of Inspection for the provision of meteorological and PM data, N. Kropf, N. Lang-Yona, and S. Lelieveld for technical assistance, N. Bothen, J. A. Huffman, and M. L. Pöhlker for helpful discussion, the Max Planck Society (MPG), the Horizon 2020 AC-

TRIS project (grant agreement No 654109) and its provision of transnational access funds, and the Deutsche Forschungsgemeinschaft (DFG, FR3641/1-2, FOR 1525 INUIT) for financial support.



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
