# Peer review of "Bioaerosols and atmospheric ice nuclei in a Mediterranean dryland: Community changes related to rainfall"

_Biogeosciences, 2021_

## Referee Comment (RC2)

**Review BG-2021-187**

Tang et al. present a paper that describes the biodiversity of samples collected in Cyprus during the INUIT 2016 campaign and its variations during or after rain events. They also measured the Ice Nucleation activity in these samples, and particularly related to a biological origin.

This work represents a rather large set of data, in general the experimental section is well detailed, the figures are numerous and informative, the paper is well presented and written. The description of bio-aerosol communities is still sparse in the literature, particularly in the Mediterranean region, the data reported here are thus valuable for the scientific community and is certainly worth publishing.

However, I have a few comments.

**Aerosol sampling**: (P3 lines 61-75)

Aerosols were collected on filters every day, this means that filters were exposed to dry air or to rain drops when rain events eventually occurred. My question is the following: do you consider that samples collected during rain events can be considered as air samples or as rain samples (or a mixture of both)? This is quite important as it can modulate the final discussion. For instance, the biodiversity or IN activity of the specific samples impacted by the rain events should be compared with those of rain rather that air samples described in the literature.

Could the authors comment on these points?

**Bacterial communities** (P12 Fig 5)

In general, whatever are the atmospheric samples (rain, snow, cloud, air) the main described phyla are Proteobacteria, Fimicutes, Actinobacteria and Bacteroidetes, with usually a prevalence of Proteobacteria (and particularly Alpha- and Gamma-Proteobacteria) ( see for review A.-M. Delort, P. Amato, Editors Microbiology of Aerosols, 2017, Wiley, (ISBN: 9781119132288).

In this work, Probacteria are not the most abundant

In addition, a number of papers report an increase the presence of some genera such as *Pseudomonas* (belonging to Gamma-Proteobacteria) in rain samples, again this is not observed in CY20 and CY23 samples that experienced rain events. (Monteil et al. Features of air masses associated with the deposition of *Pseudomonas syringae* and *Botrytis cinerea* by rain and snowfall, The ISME Journal 8, 2290–2304, 2014, https://doi.org/10.1038/ismej.2014.55 )

Could the authors discuss these results in more details?

**IN activity** (P13)

Although the IN activity measurement is correct, including experiments where heating and filtering are used, I have a concern regarding the lack of data related to the number of cells present in the sample. The IN activities of the different samples are compared but we have no information about the total number of cells, the number of bacteria or fungi. Could the authors add this important information?

In the absence of these data is it very difficult to compare the biodiversity (which is just a qualitative aspect) and the IN activity values.

**Discussion:**

Some papers are well documented on the comparison of biodiversity or IN activity of air, rain and other atmospheric samples. I suggest that the authors should compare their results with those of theses paper (see references bellow). Also I the authors do not identify major microorganisms known as efficient ice nucleators, they propose some unidentified organisms or other types of molecules responsible for the IN activity measured in their sample. However this is highly speculative there is no the strong evidence for that. In addition due to my concern about the IN section (see above), I suggest that the authors highly modulate these conclusions,

Rachel A. Moore, Regina Hanlon, Craig Powers, David G. Schmale, III and Brent C. Christner Scavenging of Sub-Micron to Micron-Sized Microbial Aerosols during Simulated Rainfall Atmosphere 2020, 11, 80; doi:10.3390/atmos11010080

References of interest:
Nora Els, Catherine Larose., Kathrin Baumann-Stanzer, Romie Tignat-Perrier, Christoph Keuschnig, Timothy M. Vogel, Birgit Sattler.
Microbial composition in seasonal time series of free tropospheric air and precipitation reveals community separation. Aerobiologia, 35, 671–701 (2019) https://doi.org/10.1007/s10453-019-09606-x

Regina Hanlon, Craig Powers, Kevin Failor, Caroline L. Monteil, Boris A. Vinatzer, and David G. Schmale III Microbial ice nucleators scavenged from the atmosphere during simulated rain events. Atmospheric Environment, 163, 2017, 182-189.
https://doi.org/10.1016/j.atmosenv.2017.05.030

KEN A. AHO, CAROLYN F. W EBER, BRENT C. CHRISTNER, BORIS A. VINATZER, CINDY E. MORRIS, RACHEL JOYCE, KEVIN C. FAILOR, JASON T. WERTH, AURORA L. H. BAYLESS-EDWARDS, AND DAVID G. SCHMALE III. Spatiotemporal patterns of microbial composition and diversity in precipitation. Ecological Monographs, 0(0), 2019, e01394

Glwadys Pouzet, Elodie Peghaire, Maxime Aguès, Jean-Luc Baray, Franz Conen and Pierre Amato. Atmospheric Processing and Variability of Biological Ice Nucleating Particles in Precipitation at Opme, France. Atmosphere 2017, 8, 229; doi:10.3390/atmos811022

---

## Author Response (AR1)

**Response to referee #2 comments on the manuscript BG-2021-187**

(Referee comments in italics, the responses in plain font)

*Referee 2: Tang et al. present a paper that describes the biodiversity of samples collected in Cyprus during the INUIT 2016 campaign and its variations during or after rain events. They also measured the Ice Nucleation activity in these samples, and particularly related to a biological origin. This work represents a rather large set of data, in general the experimental section is well detailed, the figures are numerous and informative, the paper is well presented and written. The description of bio-aerosol communities is still sparse in the literature, particularly in the Mediterranean region, the data reported here are thus valuable for the scientific community and is certainly worth publishing. However, I have a few comments.*

Response: We thank the referee for the positive feedback and the recommendation for publication. We followed up on her/his comments and suggestions, revising and clarifying the manuscript as detailed below.

*Referee 2: Aerosol sampling: (P3 lines 61-75)*
*Aerosols were collected on filters every day, this means that filters were exposed to dry air or to rain drops when rain events eventually occurred. My question is the following: do you consider that samples collected during rain events can be considered as air samples or as rain samples (or a mixture of both)? This is quite important as it can modulate the final discussion. For instance, the biodiversity or IN activity of the specific samples impacted by the rain events should be compared with those of rain rather than air samples described in the literature.*
*Could the authors comment on these points?*

Response: We consider our samples to be air samples. We collected the aerosols (i.e., total suspended particles) on filters with a High-Volume sampler. The filters were not exposed to the raindrops as they were shielded from the direct impact of precipitation due to the design of the sampler and its head. Thus, the measurements refer to particles that were not embedded in or scavenged by raindrops at the time of sampling. For clarification, we added the following sentence in the manuscript:

"Note that the measurements refer to atmospheric particles that were not embedded in or scavenged by raindrops. "

*Referee 2: Bacterial communities (P12 Fig 5)*
*In general, whatever are the atmospheric samples (rain, snow, cloud, air) the main described phyla are Proteobacteria, Fimicutes, Actinobacteria and Bacteroidetes, with usually a prevalence of Proteobacteria (and particularly Alpha- and Gamma-Proteobacteria) ( see for review A.-M. Delort, P. Amato, Editors Microbiology of Aerosols, 2017, Wiley, (ISBN: 9781119132288). In this work, Probacteria are not the most abundant.*

Response: We agree with the referee that Proteobacteria were found to be among the most abundant bacterial groups in many studies. A general dominance of Proteobacteria over the other phyla in air, however, is not supported by the available literature, and variations related to locations, seasons, and dust events are reported (e.g., Fierer et al., 2008, Bowers et al., 2011, Jeon et al., 2011, Franzetti et al., 2011, Bertolini et al., 2013, Mazar et al., 2016, Gat et al., 2017, Els et al., 2019, Mescioglu et al., 2019, Uetake et al., 2020).

We found high read proportions of Firmicutes, which agrees well with Mazar et al., 2016, and Mescioglu et al., 2019, who also found a high relative abundance of Firmicutes in aerosol samples from the Mediterranean area. We added the following sentence in the manuscript and added Mazar et al., 2016 to the reference list.

"This result agrees well with Mazar et al. (2016) and Mescioglu et al. (2019) who also found high relative abundances of Firmicutes in aerosol samples from the Mediterranean area.

Moreover, air quality is not only influenced by long-range transported particles but also by local emission sources. As Firmicutes are not only present in soil and dust but also in the intestines of larger organisms as part of the gut microbiome, local emissions of Firmicutes by the animal waste treatment company close to the sampling station could explain not only the finding of methanogens but also this result. We had pointed this out for methanogens in the Archaea section but not for Firmicutes, so we thank the referee for her/his comment and added the following sentences in the manuscript:

"An animal waste treatment company close to the sampling station might be an important local emission source for Firmicutes. "

*Referee 2: In addition, a number of papers report an increase the presence of some genera such as Pseudomonas (belonging to Gamma-Proteobacteria) in rain samples, again this is not observed in CY20 and CY23 samples that experienced rain events. (Monteil et al. Features of air masses associated with the deposition of Pseudomonas syringae and Botrytis cinerea by rain and snowfall, The ISME Journal 8, 2290–2304, 2014, https://doi.org/10.1038/ismej.2014.55 )*
*Could the authors discuss these results in more details?*

Response: In this instance, the referee misunderstands the samples we analyzed. Our samples CY20 and CY23 were air samples that were not exposed to the raindrops. In response to the referee's first comment, we have clarified this in the manuscript as explained above.

*Referee 2: IN activity (P13)*
*Although the IN activity measurement is correct, including experiments where heating and filtering are used, I have a concern regarding the lack of data related to the number of cells present in the sample. The IN activities of the different samples are compared but we have no information about the total number of cells, the number of bacteria or fungi. Could the authors add this important information?*
*In the absence of these data is it very difficult to compare the biodiversity (which is just a qualitative aspect) and the IN activity values.*

Response: We agree with the referee that cell and spore numbers would indeed be a great result of the study, but unfortunately these data are not available. We cannot obtain these data from our filter samples due to the kind of sample. The sampling procedure (e.g., filter type) has been optimized for bioaerosol DNA sequencing, one of the main purposes of this study. While the samples work fine for IN and a range of other measurements, earlier experiments with similar samples (same filter type, etc.) and cell counting by microscopy, however, were not successful. One reason seems to be the type of filter, which is a glass fiber filter. Glass fiber filters are fibrous filters that trap the particles in a network of randomly oriented fibers. They are widely used for bioaerosol collection and DNA analyses. One advantage of these filters is, that they can be decontaminated by baking at 300 - 500 °C before sampling, an

essential requirement when the sampling purpose is DNA analysis. For microscopic investigations, however, membrane filters are recommended (A.-M. Delort, P. Amato, Editors Microbiology of Aerosols, 2017, Wiley, (ISBN: 9781119132288)), but membrane filters do on the one hand often not work with our High-Volume sampler and on the other hand most of them cannot be baked at the high temperatures required for DNA decontamination. Alternative methods such as quantitative PCR to estimate concentrations of selected airborne species could also not be applied due to sample and DNA extract limitation.

In view of IN activity and the comparison with biodiversity, however, we don´t think that the number of cells and spores is essential. Only a limited number of fungal and bacterial species is known to exhibit ice nucleation activity. For IN fungi we know that most of them possess ice nucleation active macromolecules detached from the cells/spores (e.g., Fröhlich-Nowoisky et al., 2012, O´Sullivan et al., 2015, Kunert et al., 2019). For bacteria, it is well known that not every cell of an IN active strain is IN active (Lindow et al., 1982) so that the fraction of cells exhibiting IN activity is very low with respect to the total cell numbers of that species present. Moreover, fragments of IN bacteria can also be IN active. So even if we could provide an estimate of the airborne concentrations of spores and cells of a given species, this would not necessarily provide the number that could influence ice formation in the atmosphere.

*Referee 2: Discussion: Some papers are well documented on the comparison of biodiversity or IN activity of air, rain and other atmospheric samples. I suggest that the authors should compare their results with those of theses paper (see references bellow). Also I the authors do not identify major microorganisms known as efficient ice nucleators, they propose some unidentified organisms or other types of molecules responsible for the IN activity measured in their sample. However this is highly speculative there is no the strong evidence for that. In addition due to my concern about the IN section (see above), I suggest that the authors highly modulate these conclusions,*

*Rachel A. Moore, Regina Hanlon, Craig Powers, David G. Schmale, III and Brent C. Christner Scavenging of Sub-Micron to Micron-Sized Microbial Aerosols during Simulated Rainfall Atmosphere 2020, 11, 80; doi:10.3390/atmos11010080*

*References of interest:*
*Nora Els, Catherine Larose., Kathrin Baumann-Stanzer, Romie Tignat-Perrier, Christoph Keuschnig, Timothy M. Vogel, Birgit Sattler.*
*Microbial composition in seasonal time series of free tropospheric air and precipitation reveals community separation. Aerobiologia, 35, 671–701 (2019)*
*https://doi.org/10.1007/s10453- 019-09606-x*

*Regina Hanlon, Craig Powers, Kevin Failor, Caroline L. Monteil, Boris A. Vinatzer, and David G. Schmale III Microbial ice nucleators scavenged from the atmosphere during simulated rain events. Atmospheric Environment, 163, 2017, 182-189.*
*https://doi.org/10.1016/j.atmosenv.2017.05.030*

*KEN A. AHO, CAROLYN F. W EBER, BRENT C. CHRISTNER, BORIS A. VINATZER, CINDY E. MORRIS, RACHEL JOYCE, KEVIN C. FAILOR, JASON T. WERTH, AURORA L. H. BAYLESS-EDWARDS, AND DAVID G. SCHMALE III. Spatiotemporal patterns of microbial composition and diversity in precipitation. Ecological Monographs, 0(0), 2019, e01394*

*Glwadys Pouzet, Elodie Peghaire, Maxime Aguès, Jean-Luc Baray, Franz Conen and Pierre Amato. Atmospheric Processing and Variability of Biological Ice Nucleating Particles in Precipitation at Opme, France. Atmosphere 2017, 8, 229; doi:10.3390/atmos811022*

Response: We think that the referee may have misunderstood what kind of samples we analyzed. As pointed out by the referee, there is very well-established literature showing that bacteria and other particles are scavenged from the atmosphere by rain events. Indeed, analysis of real precipitation samples provides information about particles involved in droplet formation in the clouds as well as from particles scavenged by falling raindrops. We analyzed particles that were not embedded in or scavenged by raindrops as the filters were not exposed to the rain during sampling. We thank the reviewer for the suggested references. From most of them, however, for the reasons mentioned above we were not able to extract specific results that can be compared with our data set obtained from air samples. Aho et al., 2019, Hanlon et al., 2017 and Moore et al. 2020 analyzed precipitation samples from real or simulated precipitation events but did not determine or report the composition of bioparticles in the air.

The referee also suggested comparing with Els et al., 2019 who found for tropospheric air that the airborne community is distinct from the communities in precipitation samples. As we did not analyze precipitation samples, we cannot say if this would be similar in Cyprus boundary layer air and rain. This is certainly a motivation for future investigations.

We added the following sentence in the conclusion section in the manuscript and added Els et al., 2019 to the reference list:

"The bioaerosol communities need to be compared with the communities of cloud water and precipitation samples, as differences in the community compositions can provide insights into potential selection processes of specific bioaerosols and thus may provide hints to potential but previously unknown IN active microorganisms (Els et al., 2019)."

Regarding the detection of ice-nucleating microorganisms, Els et al., 2019 reported more than 50 bacterial genera and read proportions of up to ~5 % for air samples, much higher values than we found in our study. This might be explained by the different sampling location and methods used. Els et al., 2019 included a wider range of genera/species in their analysis for IN-active bacteria. However, we cannot find literature evidence for some of the most abundant IN active species such as Caulobacter, Salmonella, Rhodobacter, and Variovorax that are listed as IN active in Els et al. 2019 and there are no references provided. In our study, we refrained from including bacterial genera for which we had no reference and evidence of IN activity. We only included genera and species for which freezing characteristics and references are available. Our data suggest that there are unknown sources of atmospheric bio-IN. It is also well supported by the cited literature that these can be intact cells, cell fragments, or cell-free IN macromolecules. The identification of the so far unknown cell-free IN macromolecules from fungi and plant pollen is challenging and more laboratory work is necessary to be able to quantify them and to address their importance in the atmosphere. Our work will hopefully motivate experiments and observations in the future.

The last reference suggested is Pouzet et al., 2017, who measured IN concentrations of PM10 and precipitation samples collected in France at the Puy de Dôme station (1465 m a.s.l.). The IN concentrations of the aerosol samples show large variations and higher initial freezing temperatures than we found for the Cyprus air samples. The curves surprisingly end at temperatures between ~-8 - -14 °C, although one would expect for all samples IN in the entire

measured temperature range which was until -14°C. This allows a rough comparison with our data, but note that there is no temperature that gives a concentration for all samples of both studies. In Cyprus air, we found between 0.1 and 0.01 IN per Liter of air at -14/-15 °C, while in the French study between ~0.02 and 1 IN per m³ of air were measured at -8 °C. Unfortunately, the study does not indicate which of the aerosol samples were collected during or after a rain event so that we cannot find out if a similar trend as we see in Cyprus is reflected in their data.

**References:**

Bowers, R. M., Sullivan, A. P., Costello, E. K., Collett, J. L., Knight, R. and Fierer, N.: Sources of bacteria in outdoor air across cities in the midwestern United States., Appl. Environ. Microbiol., 77(18), 6350–6356, doi:10.1128/AEM.05498-11, 2011.

Els, N., Larose, C., Baumann-Stanzer, K., Tignat-Perrier, R., Keuschnig, C., Vogel, T. M. and Sattler, B.: Microbial composition in seasonal time series of free tropospheric air and precipitation reveals community separation, Aerobiologia (Bologna)., 0123456789, doi:10.1007/s10453-019-09606-x, 2019.

Fierer, N., Liu, Z., Rodríguez-Hernández, M., Knight, R., Henn, M. and Hernandez, M. T.: Short-term temporal variability in airborne bacterial and fungal populations., Appl. Environ. Microbiol., 74(1), 200–7, doi:10.1128/AEM.01467-07, 2008.

Franzetti, A., Gandolfi, I., Gaspari, E., Ambrosini, R. and Bestetti, G.: Seasonal variability of bacteria in fine and coarse urban air particulate matter, Appl. Microbiol. Biotechnol., 90(2), 745–753, doi:10.1007/s00253-010-3048-7, 2011.

Fröhlich-Nowoisky, J., Hill, T. C. J., Pummer, B. G., Yordanova, P., Franc, G. D. and Pöschl, U.: Ice nucleation activity in the widespread soil fungus Mortierella alpina, Biogeosciences, 12(4), 1057–1071, doi:10.5194/bg-12-1057-2015, 2015.

Gat, D., Mazar, Y., Cytryn, E. and Rudich, Y.: Origin-Dependent Variations in the Atmospheric Microbiome Community in Eastern Mediterranean Dust Storms, Environ. Sci. Technol., 51(12), 6709–6718, doi:10.1021/acs.est.7b00362, 2017.

Jeon, E. M., Kim, H. J., Jung, K., Kim, J. H., Kim, M. Y., Kim, Y. P. and Ka, J.-O.: Impact of Asian dust events on airborne bacterial community assessed by molecular analyses, Atmos. Environ., 45(25), 4313–4321, doi:10.1016/j.atmosenv.2010.11.054, 2011.

Kunert, A. T., Pöhlker, M. L., Tang, K., Krevert, C. S., Wieder, C., Speth, K. R., Hanson, L. E., Morris, C. E., Schmale III, D. G., Pöschl, U. and Fröhlich-Nowoisky, J.: Macromolecular fungal ice nuclei in Fusarium: effects of physical and chemical processing, Biogeosciences, 16(23), 4647–4659, doi:10.5194/bg-16-4647-2019, 2019.

Lindow, S. E., Hirano, S. S., Barchet, W. R., Arny, D. C. and Upper, C. D.: Relationship between Ice Nucleation Frequency of Bacteria and Frost Injury, Plant Physiol., 70(4), 1090–1093, doi:10.1104/pp.70.4.1090, 1982.

Mazar, Y., Cytryn, E., Erel, Y. and Rudich, Y.: Effect of Dust Storms on the Atmospheric Microbiome in the Eastern Mediterranean, Environ. Sci. Technol., 50(8), 4194–4202, doi:10.1021/acs.est.5b06348, 2016.

Mescioglu, E., Rahav, E., Belkin, N., Xian, P., Eizenga, J., Vichik, A., Herut, B. and Paytan, A.: Aerosol Microbiome over the Mediterranean Sea Diversity and Abundance, Atmosphere (Basel)., 10(8), 440, doi:10.3390/atmos10080440, 2019.

O′Sullivan, D., Murray, B. J., Ross, J. F., Whale, T. F., Price, H. C., Atkinson, J. D., Umo, N. S. and Webb, M. E.: The relevance of nanoscale biological fragments for ice nucleation in clouds, Sci. Rep., 5, 8082, doi:10.1038/srep08082, 2015.

Uetake, J., Hill, T. C. J., Moore, K. A., DeMott, P. J., Protat, A. and Kreidenweis, S. M.: Airborne bacteria confirm the pristine nature of the Southern Ocean boundary layer, Proc. Natl. Acad. Sci., 202000134, doi:10.1073/pnas.2000134117, 2020.